# Clinical Significance of Gamma-Glutamyltranspeptidase Combined with Carbohydrate-Deficient Transferrin for the Assessment of Excessive Alcohol Consumption in Patients with Alcoholic Cirrhosis

**DOI:** 10.3390/medicines8070039

**Published:** 2021-07-19

**Authors:** Akihiko Shibamoto, Tadashi Namisaki, Junya Suzuki, Takahiro Kubo, Satoshi Iwai, Fumimasa Tomooka, Soichi Takeda, Yuki Fujimoto, Masahide Enomoto, Koji Murata, Takashi Inoue, Koji Ishida, Hiroyuki Ogawa, Hirotetsu Takagi, Daisuke Kaya, Yuki Tsuji, Takahiro Ozutsumi, Yukihisa Fujinaga, Masanori Furukawa, Norihisa Nishimura, Yasuhiko Sawada, Koh Kitagawa, Shinya Sato, Hiroaki Takaya, Kosuke Kaji, Naotaka Shimozato, Hideto Kawaratani, Kei Moriya, Takemi Akahane, Akira Mitoro, Hitoshi Yoshiji

**Affiliations:** 1Department of Gastroenterology, Nara Medical University, 840 Shijo-cho, Kashihara, Nara 634-8522, Japan; a-shibamoto@naramed-u.ac.jp (A.S.); suzukij@naramed-u.ac.jp (J.S.); kubotaka@naramed-u.ac.jp (T.K.); satoshi181@naramed-u.ac.jp (S.I.); tomooka@naramed-u.ac.jp (F.T.); souitit@naramed-u.ac.jp (S.T.); yukifuji@naramed-u.ac.jp (Y.F.); masahidee@naramed-u.ac.jp (M.E.); muratak@naramed-u.ac.jp (K.M.); ishidak@naramed-u.ac.jp (K.I.); ogawah@naramed-u.ac.jp (H.O.); htakagi@naramed-u.ac.jp (H.T.); kayad@naramed-u.ac.jp (D.K.); tsujih@naramed-u.ac.jp (Y.T.); ozutaka@naramed-u.ac.jp (T.O.); fujinaga@naramed-u.ac.jp (Y.F.); furukawa@naramed-u.ac.jp (M.F.); nishimuran@naramed-u.ac.jp (N.N.); yasuhiko@naramed-u.ac.jp (Y.S.); kitagawa@naramed-u.ac.jp (K.K.); shinyasato@naramed-u.ac.jp (S.S.); htky@naramed-u.ac.jp (H.T.); kajik@naramed-u.ac.jp (K.K.); shimozato@naramed-u.ac.jp (N.S.); kawara@naramed-u.ac.jp (H.K.); moriyak@naramed-u.ac.jp (K.M.); stakemi@naramed-u.ac.jp (T.A.); mitoroak@naramed-u.ac.jp (A.M.); yoshijih@naramed-u.ac.jp (H.Y.); 2Institute for Clinical and Translational Science, Nara Medical University Hospital, 840 Shijo-cho, Kashihara, Nara 634-8522, Japan; tk-inoue@naramed-u.ac.jp

**Keywords:** chronic excessive alcohol consumption, prediction accuracy, carbohydrate-deficient transferrin, gamma-glutamyltranspeptidase, alcoholic cirrhosis

## Abstract

**Background:** This study aimed to compare the diagnostic performance of carbohydrate-deficient transferrin (CDT) and gamma-glutamyltranspeptidase (γ-GTP) to assess the single and combined benefits of these biological markers for the detection of chronic excessive alcohol consumption in patients with alcoholic cirrhosis. **Methods:** Biological markers were determined in blood samples from patients with alcoholic cirrhosis (drinking group, *n* = 35; nondrinking group, *n* = 81). The prediction accuracy of %CDT alone, γ-GTP alone, and their combination for the detection of excessive alcohol consumption was determined in patients with alcoholic cirrhosis. **Results:** Serum total bilirubin, alanine aminotransferase, aspartate aminotransferase, γ-GTP, and alkaline phosphatase levels and %CDT were significantly higher and serum albumin levels were significantly lower in the drinking group than in the nondrinking group. The combination of %CDT and γ-GTP compared with %CDT or γ-GTP alone showed a higher prediction accuracy. The combination of %CDT and γ-GTP exhibited a higher specificity than γ-GTP alone. However, in terms of sensitivity, no significant difference was found between single or combined markers. **Conclusions:** The combination of %CDT and γ-GTP is considered a useful biomarker of chronic excessive alcohol consumption in patients with alcoholic cirrhosis.

## 1. Introduction

Liver cirrhosis (LC) represents the irreversible, advanced stage of various chronic liver diseases and has become one of the leading causes of morbidity and mortality worldwide, including in Asia [1]. A recent study showed that the Asia–Pacific region accounts for 34% of global deaths caused by LC [2]. Alcohol-related liver disease (ARLD) and hepatitis C virus (HCV) are currently the leading causes of LC in developed countries, including Japan [3,4]. The ratio of HCV-associated LC has been reduced with the increased prevalence of nonviral LC due to recent advances in the development of antiviral drugs against HCV [5]. Among nonviral etiologies, the prevalence of alcoholic cirrhosis was found to be increased from 13.7% to 24.9% in a nationwide Japanese survey in 2018 [4]. ARLD is characterized by liver injury induced by alcohol abuse and eventually develops in several stages, beginning with hepatic steatosis, and, in some individuals, developing gradually into alcoholic hepatitis, followed by alcoholic liver fibrosis, with up to 20% of cases progressing to alcoholic cirrhosis [6]. Given that there are few programs for the early detection of ARLD, patients with ARLD are diagnosed at more advanced disease stages than those with viral-associated liver disease. Therefore, alcohol consumption should be assessed correctly to prevent severe liver damage and irreversible liver failure.

Urine, blood, or hair tests are used as alcohol abuse biomarkers to assess alcohol consumption and monitor alcohol abstinence. Gamma-glutamyltranspeptidase (γ-GTP) is the most common traditional biomarker of excessive alcohol intake. γ-GTP is more accurate than other traditional markers [e.g., alanine aminotransferase (ALT), aspartate aminotransferase (AST), and mean corpuscular volume] for alcohol-induced liver injury [7]. The Food and Drug Administration has approved the measurement of carbohydrate-deficient transferrin (CDT) as a blood test to assess excessive alcohol and monitor alcohol consumption. CDT is generated as a result of alcohol inhibition of transferrin glycosylation in response to chronic alcohol abuse, and its levels are measured using a commercially available test in Western countries [8]. Some patients with alcoholic cirrhosis continue drinking alcohol, despite knowing that once cirrhosis develops the risk of dying is much higher if they continue drinking. Several studies have shown that the use of test combinations such as γ-GTP and CDT can improve the prediction accuracy of excessive ethanol consumption in cases related to alcohol-related liver diseases [9,10,11]. CDT is a marker of alcohol intake that is used for detecting or monitoring alcohol use disorders (AUDs) regardless of sex [12,13]. Whereas absolute CDT values are sex-dependent, %CDT is not [14]. Furthermore, no significant association between sex and increased %CDT has been found [15]. CDT refers to changes in the microheterogeneity of the iron transport glycoprotein transferrin that are observed after excessive alcohol consumption. Transferrin molecules in the blood usually contain several carbohydrate components. Transferrin usually adheres to different carbohydrate components. Ethanol and acetaldehyde inhibit the enzymes responsible for the addition of the carbohydrate side chains and induce the production of sialidase, which cleaves the terminal sialic acid residues from the side chains. Therefore, in patients who are chronic heavy drinkers, the number of carbohydrate components in each transferrin molecule decreases, thereby elevating the level of CDT [16]. However, CDT levels have not been correlated with fibrotic indices or even a decrease in advanced liver fibrosis [17]. This study aimed to explore the predictive accuracy of %CDT and γ-GTP alone or in combination for the management of chronic excessive alcohol intake in patients with alcoholic cirrhosis.

## 2. Materials and Methods

### 2.1. Patients

This retrospective observational study enrolled a single-center cohort between February 2015 and December 2019 at Nara Medical University Affiliated Hospital. In total, 116 consecutive patients aged 20–80 years with alcoholic cirrhosis were enrolled, and all patients continued excessive drinking until the diagnosis of alcoholic cirrhosis. Some patients with alcoholic cirrhosis stopped drinking, but others continued drinking alcohol, even after the development of cirrhosis. The patients were categorized into two groups: drinking (*n* = 35), defined as patients who drank ≥60 g and ≥40 g of pure alcohol per day for men and women, respectively, over a minimum of two weeks before blood samples were collected [18]; and nondrinking (*n* = 81), defined as patients who drank pure alcohol at <20 g/day (men) and 10 g/day (women) up to five times per week [19] (Figure 1). Of the 81 patients, nine remained abstinent after they were diagnosed with cirrhosis. We calculated the average amount of alcohol intake per day based on the self-reported measures of alcohol consumption as reported by patients to their general practitioner or practice nurse [20].

The collection of blood samples was conducted at least four weeks after the diagnosis of alcoholic cirrhosis, and an average abstinence period was 6.7 months in the nondrinking group. The diagnosis of LC was made based on the combination of laboratory tests such as total bilirubin, albumin and prothrombin time (PT), clinical and imaging features, or liver histology findings. Patients were subsequently categorized into three groups (Child–Pugh classes A, B, or C) according to Pugh’s criteria (https://www.merckmanuals.com/medical-calculators/ChildTurPuScore.htm, accessed on 3 July 2021). Patients with hepatocellular carcinoma, extrahepatic cancers, infectious diseases, and concomitant liver disease (i.e., chronic hepatitis B infection, hepatitis C infection, primary biliary cholangitis, and autoimmune hepatitis) were excluded. The primary outcome is the prediction efficacy of %CDT combined with γ-GTP in detecting excessive alcohol consumption in patients with alcoholic cirrhosis. %CDT, γ-GTP, ALT, and AST were measured on the diagnostic day. All procedures conducted in this study involving human participants were based on the ethical standards of the Medical Ethics Committee of Nara Medical University (Nara-med, 052-0141 approved on 15 December 2014) and the 1964 Helsinki Declaration and its later amendments. Written informed consent for blood specimen collection was obtained from all 116 patients enrolled. All study patients voluntarily provided informed written consent prior to study enrollment.

### 2.2. CDT and Biochemical Parameter Measurements

Serum CDT% levels were measured with an N Latex CDT direct immunonephelometric assay (Siemens Healthcare Diagnostics, Marburg, Germany), a monoclonal antibody-based direct immunoassay that identifies the structure of transferrin glycoforms with a shortage of one or two complete N-glycans that correspond to disialo-, monosialo-, and asialo-transferrins (CDT glycoforms) [21]. Concomitant measurements of total transferrin with a direct immunonephelometric assay using polyclonal antibodies (Siemens Healthcare Diagnostics, Japan), conducted on the same instrument, can automatically determine relative CDT concentrations expressed as the percentage of the total transferrin concentration (%CDT). The Child–Pugh score grades the liver disease severity, consisting of the presence of ascites and encephalopathy, albumin (ALB) and total bilirubin (T-Bil) levels, and prothrombin time% (PT%), with the following classification: Child–Pugh class A, 5–6 points; class B, 7–9 points; and class C, 10–15 points.

### 2.3. Statistical Analysis

Normal (ALB, PT%, platelet, AST, ALP, %CDT, IgA, transferrin, and iron levels) and non-normal (γ-GTP, ALT, and T-Bil levels) distributions of continuous variables were expressed as a mean ± standard deviation and median (interquartile range [IQR]; minimum and maximum), respectively, whereas categorical variables were presented in a contingency table. Baseline characteristics were compared between the groups using the Mann–Whitney U test for γ-GTP, ALT, and total bilirubin (nonparametric; Student’s *t*-test for other continuous variables [normally distributed variables] and Fisher’s exact test for categorical variables). Spearman’s rank correlation coefficient was used to evaluate the association between %CDT and γ-GTP.

The prediction accuracy indices of biomarkers for chronic alcohol abuse diagnosis, i.e., sensitivity, specificity, and area under the curve (AUC), were evaluated using the receiver operating characteristic (ROC) curve. The estimation of the confidence interval of AUC and statistical comparison between the AUCs were performed using the bootstrap method. Akaike’s information criterion was used for evaluating the relative goodness-of-fit quality of a statistical model for a given set of data, and the McNemar test was employed to compare the models in terms of sensitivity and specificity. All analyses were performed using R Ver.4.0.2 (The R foundation for Statistical Computing, Vienna, Austria). Two-sided *p* < 0.05 was considered statistically significant.

## 3. Results

### 3.1. Baseline Physical and Biochemical Characteristics

Table 1 presents the physical characteristics of patients with alcoholic cirrhosis. Among 116 patients with alcoholic cirrhosis, 108 (93.1%) were men and eight (6.9%) were women, with a mean age of 65.5 ± 11.8 years. No significant differences in the body mass index (BMI) were observed between the drinking and nondrinking groups. The patients were classified according to their BMI into malnutrition (BMI < 18.5 kg/m^2^), normal weight (18.5 kg/m^2^ ≤ BMI < 25 kg/m^2^), overweight (25 kg/m^2^ ≤ BMI < 30 kg/m^2^), and obesity (30 kg/m^2^ ≤ BMI). The prevalence rates of malnutrition, normal weight, overweight, and obesity were 7.8% (9/116), 57.8% (67/116), 24.1% (28/116), and 10.3% (12/116), respectively, in all patients. The prevalence rates of these categories were 8.6% (3/35), 54.3% (19/35), 31.4% (11/35), and 5.7% (2/35), respectively, in the drinking group and 7.4% (6/81), 59.3% (48/81), 21.0% (17/81), and12.3% (10/81), respectively, in the nondrinking group (Figure 2).

No significant difference was noted between the two groups in the distribution of patients according to Child–Pugh class (Table 2). the serum ALB levels were significantly lower in the drinking than in the nondrinking group (*p* = 0.019). Serum T-Bil, AST, ALT, γ-GTP, alkaline phosphatase (ALP), and %CDT levels were significantly higher in the drinking group than in the nondrinking group. The ALB level was significantly higher in the nondrinking group than in the drinking group. No significant differences in PLT count, PT%, and IgA, transferrin, and iron levels were noted between the two groups. Furthermore, no significant differences were observed in %CDT values among Child–Pugh classes A, B, and C (Appendix A).

### 3.2. Correlation between the %CDT and γ-GTP

The correlation between %CDT and γ-GTP levels was 0.34 (*p* < 0.001) (Figure 3). Figure 4 shows the ROC curves for CDT and γ-GTP alone and in combination.

The predictive performance of either marker alone for the prediction of patients with alcoholic cirrhosis who continued alcohol drinking was comparable (Figure 4). The combination of the two markers significantly improved the predictive performance compared with that of %CDT or γ-GTP alone.

### 3.3. Prediction Accuracy of %CDT Alone, γ-GTP Alone, and Their Combination for Alcohol Consumption

The prediction accuracy of γ-GTP and %CDT alone for the drinking group was 79.2% (74.3% sensitivity and 70.4% specificity) and 79.0% (65.7% sensitivity and 80.2% specificity), respectively. The combination of these markers showed a higher accuracy than either of the biomarkers alone (both *p* < 0.05) (Table 3). The combination of the two markers yielded more sensitivity than either marker alone; however, the differences were not statistically significant. The combination of these markers had a significantly better specificity than γ-GTP alone but not %CDT alone (*p* = 0.021 and 0.75, respectively). A combined index based on γ-GTP and %CDT measurements had a significantly better prediction performance than γ-GTP alone at higher specificity levels.

## 4. Discussion

γ-GTP is the most frequently used biomarker for the evaluation of alcohol consumption. Among the various indirect biomarkers [22], CDT is the most specific serum biomarker of excessive alcohol consumption [23]. %CDT showed a high sensitivity but relatively low specificity. In this study, a combination of %CDT and γ-GTP was found to be a useful marker of excessive alcohol consumption in patients with alcoholic cirrhosis. Alcohol consumption biomarkers are physiological indicators of alcohol exposure and may reflect the presence of AUDs. However, the main disadvantage of CDT is its relatively low sensitivity. To the best of our knowledge, this is the first study to show that the combination of these two markers improves the prediction efficacy compared with either single marker for detecting alcoholic cirrhosis in patients who continue to drink alcohol. The response of γ-GTP to alcohol consumption varies among individuals [24]. The association between increased γ-GTP levels and alcohol consumption may not necessarily reflect a causal effect of alcohol intake on γ-GTP due to the pleiotropy of genetic variants affecting multiple traits. Several studies have shown that %CDT and γ-GTP are independent and complimentary biomarkers for the detection of chronic alcohol abuse [25,26,27]. In this study, the γ-GTP levels were within the normal range in eight patients in the drinking group. The serum CDT level can be used to differentiate between nondrinkers and excessive drinkers whose γ-GTP levels are within the normal range. Among them, three patients were found to have a %CDT higher than the cutoff values (data not shown). These findings indicate that %CDT and γ-GTP complement each other, and thus their combination can be used to identify more patients with alcoholic cirrhosis who continue to drink alcohol than when using γ-GTP alone. An improved prediction accuracy in detecting AUDs could be achieved by combining two or more alcohol markers [9,27]. The conventional manner of combining markers is to investigate whether either marker has a higher efficacy than the optimal cutoff value [9]. This could lead to a higher sensitivity, but it is correlated with a decreased specificity in predicting excessive alcohol consumption. Schwan et al. recently reported that combining γ-GTP and %CDT as independent parameters yielded a sensitivity of >90% for AUDs, whereas the specificity was 63% [28]. However, consistent with our study, %CDT had a high sensitivity but relatively low specificity. Moreover, the results showed that for AUDs a combined index based on γ-GTP and %CDT measurements was more effective in detecting alcohol dependence than using either marker alone (sensitivity and specificity of 72% and 90%, respectively, for the combination test) [23]. With a sensitivity and specificity of 83% and 95%, respectively, γ-GTP in combination with CDT appears to be an excellent biomarker for the detection of chronic alcohol abuse [27]. Differences in prediction performances between studies remain unclear, which might be partially explained by the fact that %CDT values could be affected by multiple racial and ethnic backgrounds. γ-GTP is the most frequently used biomarker for alcohol consumption. The relative level of serum CDT is currently the most specific serum biomarker of alcohol abuse [22]. These findings strengthen the hypothesis that the combined use of these two biomarkers provides complementary information and increases the predictive accuracy for chronic alcohol intake in patients with alcoholic cirrhosis.

Nevertheless, this study has several limitations that should be acknowledged. First, this was a single-center study that enrolled a relatively small number of patients with alcoholic cirrhosis without a control group of patients (*n* = 116). Second, we did not analyze %CDT and γ-GTP levels in nonalcohol (HBV, HCV, and non alcoholic steatohepatitis-related cirrhosis. Third, the patients’ reported alcohol consumption could be under- or overestimated. The biomarkers seem to complement the objective measures of self-reported alcohol consumption.

## 5. Conclusions

%CDT is a clinically useful biomarker for diagnosing AUDs, especially in patients with nonspecifically increased γ-GTP levels. Our study results confirm that %CDT in combination with γ-GTP is an effective marker for chronic alcohol consumption in patients with alcoholic cirrhosis.

## Figures and Tables

**Figure 1 medicines-08-00039-f001:**
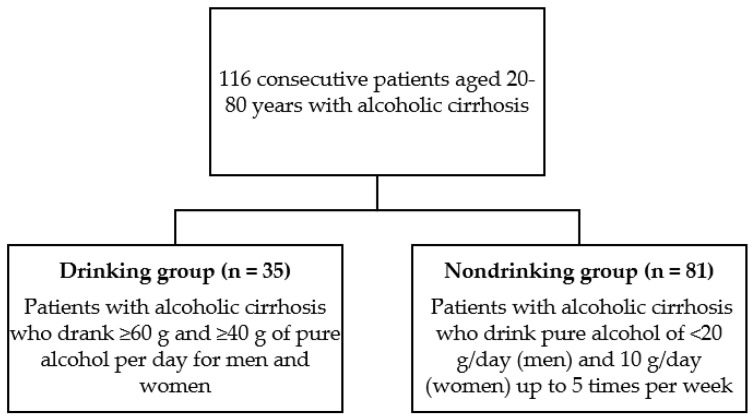
Flow chart of patient enrollment in the study.

**Figure 2 medicines-08-00039-f002:**
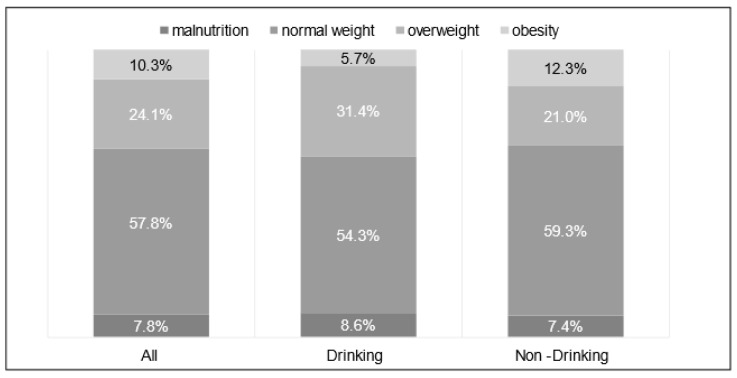
Prevalence of malnutrition, normal weight, overweight, and obesity categories.

**Figure 3 medicines-08-00039-f003:**
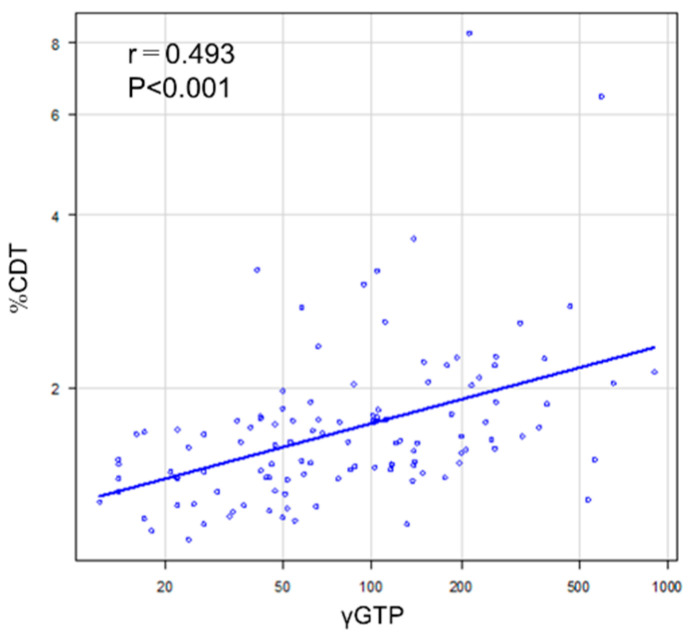
Correlation between serum carbohydrate-deficient transferrin and gammaglutamyl transferase levels.

**Figure 4 medicines-08-00039-f004:**
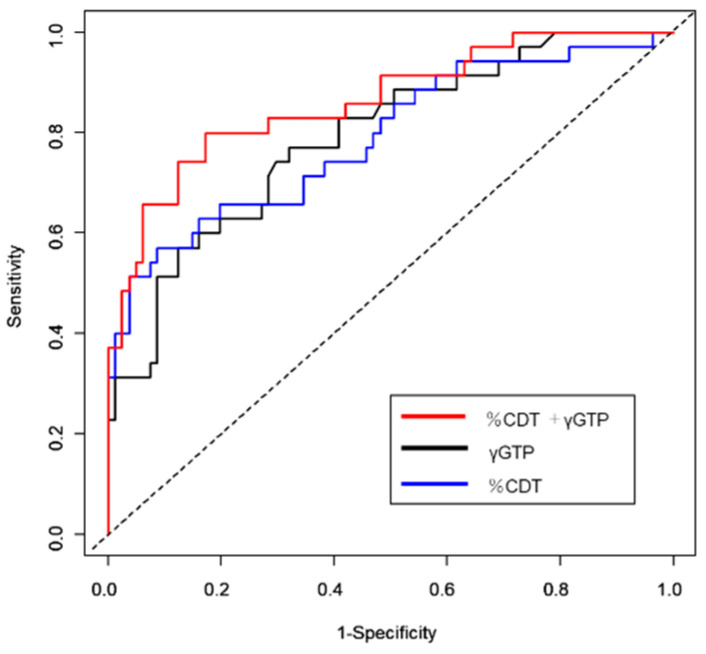
Receiver operating characteristic curves for % of carbohydrate-deficient transferrin and gammaglutamyl transferase alone and combined.

**Table 1 medicines-08-00039-t001:** Comparison of physical backgrounds of patients with alcoholic cirrhosis between the drinking and nondrinking groups.

Characteristics	All Patients (*n* = 116)	Drinking Group (*n* = 35)	Nondrinking Group (*n* = 81)	*p*-Value
Age (years)	65.5 ± 11.8	62.4 ± 11.4	66.9 ± 11.7	0.062
Male/Female	108/8	32/3	76/5	0.70
Height (cm)	165.0 ± 7.2	165.5 ± 7.1	164.7 ± 7.2	0.60
Weight (kg)	65.6 ± 13.1	65.3 ± 11.5	65.8 ± 13.7	0.83
Body mass index (kg/m^2^)	24.0 ± 4.0	23.8 ± 3.5	24.1 ± 4.2	0.65

**Table 2 medicines-08-00039-t002:** Comparison of biochemical data on alcoholic cirrhosis between the drinking and nondrinking groups.

Biochemical Data	All Patients (*n* = 116)	Drinking Group (*n* = 35)	Nondrinking Group (*n* = 81)	*p*-Value
Albumin, mean ± SD	3.7 ± 0.7	3.5 ± 0.7	3.8 ± 0.6	0.019
PT, %	71.1 ± 16.8	68.5 ± 22.8	72.2 ± 13.0	0.28
T-Bil	1.2 (0.8–1.6)	1.5 (0.95–3.3)	1.1 (0.8–1.5)	0.011
Child–Pugh classification (A/B/C)	76/29/11	56/21/4	20/8/7	0.053
Platelet	12.2 ± 5.9	12.6 ± 6.7	12.0 ± 5.4	0.67
AST	52.4 ± 42.4	81.6 ± 57.0	39.8 ± 24.9	<0.001
ALT	25 (17–45)	44 (23.5–59.5)	23 (15–34)	<0.001
γ-GTP	77.5 (42–151.3)	194 (98–319.5)	54 (33–115)	<0.001
ALP	396.2 ± 211.3	469.2 ± 275.2	365.2 ± 166.3	0.015
%CDT	1.8 ± 0.9	2.4 ± 1.4	1.5 ± 0.3	<0.001
IgA	464.1 ± 248.3	513.0 ± 287.1	443.3 ± 230.7	0.19
Transferrin	231.4 ± 68.6	220.7 ± 79.2	236.6 ± 63.0	0.35
Iron	100.4 ± 64.3	119.9 ± 72.3	91.9 ± 59.1	0.064

PT, prothrombin time; SD, standard deviation; T-Bil, total bilirubin; AST, aspartate transaminase; ALT, alanine transaminase; γ-GTP, γ-glutamyl transpeptidase; ALP, alkaline phosphatase; CDT, carbohydrate deficient transferrin; IgA, immunoglobulin A.

**Table 3 medicines-08-00039-t003:** Comparison of AUC for γ-GTP alone, %CDT alone, and their combination.

Diagnostic Model	Cut Off	Se	Sp	PPV	NVP	AIC	AUC (95% CI)
γ-GTP (IU/L)	101.5	0.743	0.704	0.520	0.864	116.1	0.792 (0.689–0.878)
%CDT	1.75	0.657	0.802	0.590	0.844	109.7	0.790 (0.696–0.878)
γGTP + %CDT	0.25	0.800	0.827	0.667	0.905	96.4	0.863 (0.779–0.930)

AUC, area under the curve; AIC, Akaike’s information criterion; Se, sensitivity; Sp, specificity; PPV, positive predictive value; NPV, negative predictive value; CI, confidence interval; γ-GTP, γ-glutamyl transpeptidase; CDT, carbohydrate-deficient transferrin.

## Data Availability

Raw data were generated from Nara Medical University Affiliated Hospital. Derived data supporting the study findings are available from the corresponding author on reasonable request.

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
