# Peer review of "Clinical Significance of Gamma-Glutamyltranspeptidase Combined with Carbohydrate-Deficient Transferrin for the Assessment of Excessive Alcohol Consumption in Patients with Alcoholic Cirrhosis"

_medicines, 2021, doi:10.3390/medicines8070039_

Round 1
Reviewer 1 Report
The revised manuscript was well-reconsidered and understandable. Of course, it is better to increase the number of cases, the contents are acceptable for publication.
Author Response
12 July 2021
Medicines
Manuscript ID: medicines-1305725
Originally titled: Clinical significance of gamma-glutamyltranspeptidase combined with carbohydrate-deficient transferrin for the assessment of excessive alcohol consumption in patients with alcoholic cirrhosis
Akihiko Shibamoto , Tadashi Namisaki * , Junya Suzuki , Takahiro Kubo , Satoshi Iwai , Fumimasa Tomooka , Soichi Takeda , Yuki Fujimoto , Masahide Enomoto , Koji Murata , Takashi Inoue , Koji Ishida , Hiroyuki Ogawa , Hirotetsu Takagi , Daisuke Kaya , Yuki Tsuji , Takahiro Ozutsumi , Yukihisa Fujinaga , Masanori Furukawa , Norihisa Nishimura , Yasuhiko Sawada , Koh Kitagawa , Shinya Sato , Hiroaki Takaya , Kosuke Kaji , Naotaka Shimozato , Hideto Kawaratani , Kei Moriya , Takemi Akahane , Akira Mitoro , Hitoshi Yoshiji
We are very grateful for the helpful comments and suggestions regarding our manuscript. We have thoroughly addressed all concerns and issues raised and have revised our manuscript accordingly. All changes in the revised manuscript are highlighted in blue We believe that the manuscript has been greatly improved, and we hope it is ready for publication in Medicines. Once again, we acknowledge that your comments have been extremely valuable in improving the quality of our manuscript. We have provided point-by-point responses to the editor’s and reviewers’ comments below.
Reviewer 2
Comments and Suggestions for Authors
The manuscript has been revised according to my previous comments, so I believe it may be published after minor corrections.
There are editorial errors in the manuscript:
the way the publication is cited in the text is incorrect
Author response:
Thank you for your valuable comments. We revised the manuscript accordingly.
Please indicate which data were normally distributed and which were non-parametric (the median should be added with the minimum and maximum)
Author response: Thank you for your valuable comments. Normal (ALB, PT%, platelet, AST, ALP, %CDT, IgA, transferrin and iron levels), and non-normal (γ-GTP, ALT, and T-Bil levels) distributions of continuous variables were expressed as mean ± standard deviation and median (interquartile range [IQR]; minimum and maximum), respectively,
We have included a description of these findings on page 4, lines 134–137.
there is no description for figure 2
Author response:
Thank you for your valuable comments. The prevalence rates of malnutrition, normal weight, overweight, and obesity were 7.8% (9/116), 57.8% (67/116), 24.1% (28/116), and 10.3% (12/116), respectively, in all patients. The prevalence rates of these categories were 8.6% (3/35), 54.3% (19/35), 31.4% (11/35), and 5.7% (2/35), respectively, in the drinking group and 7.4% (6/81), 59.3% (48/81), 21.0% (17/81), and12.3% (10/81), respectively, in the nondrinking group (Fig. 2). We have included a description of these findings on page 5, lines 161–165.
line 164-168 - the text is misaligned
Author response:
Thank you for your valuable comments. We revised the manuscript accordingly.
Table 2 - the font is too large
Author response:
Thank you for your valuable comments. We revised the Table2 accordingly.
in my opinion, the text requires a linguistic correction
Author response:
Thank you for your valuable comments. We revised the Table2 accordingly.
Reviewer 3
Author response:
Thank you for your valuable comments. We revised the manuscript accordingly.
We have included a description of these findings on page 1, lines 20–21.
Reviewer 2 Report
The manuscript has been revised according to my previous comments, so I believe it may be published after minor corrections.
There are editorial errors in the manuscript:
- the way the publication is cited in the text is incorrect
- Please indicate which data were normally distributed and which were non-parametric (the median should be added with the minimum and maximum)
- there is no description for figure 2
- line 164-168 - the text is misaligned
- Table 2 - the font is too large
- in my opinion, the text requires a linguistic correction
Author Response

(The authors gave the same response as above.)

Reviewer 3 Report
Mention of the control group in number 10 individuals is still present in the abstract
Author Response
12 July 2021
Medicines
Manuscript ID: medicines-1305725
Originally titled: Clinical significance of gamma-glutamyltranspeptidase combined with carbohydrate-deficient transferrin for the assessment of excessive alcohol consumption in patients with alcoholic cirrhosis
Akihiko Shibamoto , Tadashi Namisaki * , Junya Suzuki , Takahiro Kubo , Satoshi Iwai , Fumimasa Tomooka , Soichi Takeda , Yuki Fujimoto , Masahide Enomoto , Koji Murata , Takashi Inoue , Koji Ishida , Hiroyuki Ogawa , Hirotetsu Takagi , Daisuke Kaya , Yuki Tsuji , Takahiro Ozutsumi , Yukihisa Fujinaga , Masanori Furukawa , Norihisa Nishimura , Yasuhiko Sawada , Koh Kitagawa , Shinya Sato , Hiroaki Takaya , Kosuke Kaji , Naotaka Shimozato , Hideto Kawaratani , Kei Moriya , Takemi Akahane , Akira Mitoro , Hitoshi Yoshiji
We are very grateful for the helpful comments and suggestions regarding our manuscript. We have thoroughly addressed all concerns and issues raised and have revised our manuscript accordingly. All changes in the revised manuscript are highlighted in blue We believe that the manuscript has been greatly improved, and we hope it is ready for publication in Medicines. Once again, we acknowledge that your comments have been extremely valuable in improving the quality of our manuscript. We have provided point-by-point responses to the editor’s and reviewers’ comments below.
Reviewer 2
Comments and Suggestions for Authors
The manuscript has been revised according to my previous comments, so I believe it may be published after minor corrections.
There are editorial errors in the manuscript:
the way the publication is cited in the text is incorrect
Author response:
Thank you for your valuable comments. We revised the manuscript accordingly.
Please indicate which data were normally distributed and which were non-parametric (the median should be added with the minimum and maximum)
Author response: Thank you for your valuable comments. Normal (ALB, PT%, platelet, AST, ALP, %CDT, IgA, transferrin and iron levels), and non-normal (γ-GTP, ALT, and T-Bil levels) distributions of continuous variables were expressed as mean ± standard deviation and median (interquartile range [IQR]; minimum and maximum), respectively,
We have included a description of these findings on page 4, lines 134–137.
there is no description for figure 2
Author response:
Thank you for your valuable comments. The prevalence rates of malnutrition, normal weight, overweight, and obesity were 7.8% (9/116), 57.8% (67/116), 24.1% (28/116), and 10.3% (12/116), respectively, in all patients. The prevalence rates of these categories were 8.6% (3/35), 54.3% (19/35), 31.4% (11/35), and 5.7% (2/35), respectively, in the drinking group and 7.4% (6/81), 59.3% (48/81), 21.0% (17/81), and12.3% (10/81), respectively, in the nondrinking group (Fig. 2). We have included a description of these findings on page 5, lines 161–165.
line 164-168 - the text is misaligned
Author response:
Thank you for your valuable comments. We revised the manuscript accordingly.
Table 2 - the font is too large
Author response:
Thank you for your valuable comments. We revised the Table2 accordingly.
in my opinion, the text requires a linguistic correction
Author response:
Thank you for your valuable comments. We revised the Table2 accordingly.
Reviewer 3
Author response:
Thank you for your valuable comments. We revised the manuscript accordingly.
We have included a description of these findings on page 1, lines 20–21.
This manuscript is a resubmission of an earlier submission. The following is a list of the peer review reports and author responses from that submission.
Round 1
Reviewer 1 Report
This manuscript reports that both G-GT and %CDT can be useful as markers of heavy drinking. So far, CDT is used as a marker of habitual drinking, and is useful for screening chronic heavy drinkers and monitoring alcohol consumption and abstinence. The clinical significance of CDT is not novel. There are several criticisms
- The number of cases studied was small, so varidation is necessary.
- Most of the cases reviewed in this study were males. How about the sexual difference? Is it useful for both men and women?
- CDT levels are reported to be affected by transferrin levels, iron status, liver dysfunction, etc. What was the situation in this study?
- In some cases, elevated CDT values have been observed in the group of non-responders (those whose γ-GT values are not abnormal despite drinking a considerable amount of alcohol). Is there any clinical importance?
Reviewer 2 Report
Authors of the manuscript entitled „Clinical significance of gamma-glutamyltranspeptidase combined with carbohydrate-deficient transferrin for the assessment of excessive alcohol consumption in patients with alcoholic cirrhosis” present an interesting topic in the field of diagnosis of alcohol abuse, but major corrections are necessary.
Introduction
- The Introduction requires corrections. Authors begin the Introduction section with a description of HCV, which is incomprehensible given the title of the manuscript. There is no information in the title that the study concerns patients with HCV. Please explain and correct it.
- There is also no good justification for this study. Methods to assess excessive alcohol consumption are now available, so Authors should better substantiate the purpose of this study.
Materials and methods
- This section requires major corrections.
- HCV patients were excluded from this study, so what is the relationship with the information in the Introduction section?
- What were the inclusion criteria for the study regarding the age of the patients? Please complete this information.
- What were the inclusion criteria for the study regarding the status health of patients (e.g. cardiovascular disease, thyroid disease, etc.)? Authors should complete this.
- Authors wrote that: „Ten patients with gastric polyps or colorectal polyps were assigned as the control group”. Please explain why patients with polyps formed a control group? Shouldn't it be a group of healthy non-drinkers?
-What questionnaires were used to determine the frequency of alcohol consumption? Did the participants complete the questionnaires themselves? Authors must complete this.
- Please complete this section with an experiment scheme. This will make it easier to understand the course of the study.
- Please indicate on the basis of which laboratory tests and what results of these tests were assigned to particular groups. This is incomprehensible in the current description.
- How did Authors verify the normality of distribution of data? What tests did they use?
Results
- Table 1 – the specification „characteristics” in the first column is incorrect. Age and sex are not presented in a single table with biochemical data.
- Please complete the data with height, weight and BMI of patients and its interpretation. How many patients were malnourished, with normal body weight, and were overweight or obese?
Table 2 is in chapter 3.2, but that was described in chapter 3.1, it needs to be corrected. The specification „characteristics” in the first column is incorrect. Age and sex are not presented in a single table with biochemical data.
Discussion
- The Discussion section requires major corrections.
- Authors wrote immediately at the beginning of the discussion that „This study found that a combination of %CDT with γ-GTP is a useful marker of excessive alcohol consumption in patients with alcoholic cirrhosis.” The Discussion section does not start with the conclusions of the study. Please first assess whether the actually obtained results prove the usefulness of the marker used.
- It is incomprehensible to include the figures and table in the discussion. Figures and table should be transferred to the results. Moreover, there is no description of the figures and table.
- There is no Conclusions section.
Reviewer 3 Report
The presented paper is interesting and also can lead to the direct use of the results in practice, which is very useful, however only if authors register study in ClinicalTrials.gov as is required for clinical studies. ID should be staten in study (Then could be in another, following study removed mentioned limitations of the study).
Due to low number of enrolled patients, there is need to provide test of equality of variances, only if confirmed, t-test is appropriate.
Number of patients in control group is very low, however, still is. It's not perishable why there is conclusion (last paragraph) the study was without control group.